# Benchmarking free energy calculations: Analysis of single and double mutations across two simulation software platforms for two protein systems

Shivani Gupta[2], Qinfang Sun[1,2]*, Ronald M. Levy[1,2]*

**1** Center for Biophysics and Computational Biology, Temple University, Philadelphia, Pennsylvania, United States of America, **2** Department of Chemistry, Temple University, Philadelphia, Pennsylvania, United States of America

* qinfang.sun@temple.edu (QS); ronlevy@temple.edu (RML)

## Abstract

Mutation analysis of single and double mutations of proteins including the two widely studied proteins: Staphylococcus nuclease (S. nuclease) and T4 lysozyme, provides critical insights into their stability and fitness. Furthermore, such analysis facilitates a deeper understanding of the complex interplay between a protein's sequence, structural conformation, and functional output. Free Energy Perturbation (FEP) is an alchemical, all-atom molecular dynamics based computational approach that determines the free energy change ($\Delta\Delta G$) from wild type to mutant states. Two widely adopted software platforms used for this purpose are Schrödinger and GROMACS. We have compared the results of FEP simulations for mutations using these platforms, employing the OPLS4 force field implemented in Schrödinger, and the Amber99SB-ILDN force field implemented in GROMACS for this work. For the 38 single mutants of S. nuclease, the Pearson r between the experimental and the calculated free energy change ($\Delta\Delta G$) was 0.86 using Schrödinger and 0.87 using GROMACS. The reliability of the FEP method using the two software platforms with the specified force fields was further demonstrated by its performance on 24 single mutants of T4 lysozyme, yielding strong correlation between predicted and experimental $\Delta\Delta G$ values, with Pearson r of 0.80 for Schrödinger and 0.85 for GROMACS. Additionally, the computed folding free energy changes for 45 double mutations in S. nuclease ($\Delta\Delta G_{WT}^{AB}$) using Schrödinger correlated well with both experimental measurements (Pearson r = 0.74) and previously reported GROMACS values (Pearson r = 0.71). Correspondingly, the nonadditivities ($\delta_{WT}^{AB}$) of the double mutations derived using Schrödinger for these 45 double mutants of S. nuclease were also found to be in good agreement with the experimental values (Pearson r = 0.79), as well as with the previously reported GROMACS results (Pearson r = 0.61). A good correlation was also observed between computed values from Schrödinger and GROMACS, with a

**Data availability statement:** All relevant data are within the manuscript and its Supporting information files.

**Funding:** This study was supported by the National Institutes of Health (NIH) Grant R35 GM132090. The funder had no role in study design, data collection and analysis, decision to publish, or preparation of the manuscript.

**Competing interests:** The authors declare no competing interests.

Pearson r of 0.71 for double mutants and 0.61 for their nonadditivities. Collectively, these findings establish the efficiency, accuracy, and reliability of Schrödinger FEP+ and GROMACS in predicting mutation-induced free energy changes in S. nuclease and T4 lysozyme. The integration of FEP based computational methodologies with experimental validation provides a framework for quantifying mutation-induced changes in protein thermostability, which can be used as a tool for protein engineering and design.

## Introduction

Mutational analysis serves as a powerful tool for elucidating the complex interdependencies among protein sequence, structure, and function [1]. By systematically assessing the impact of specific amino acid substitutions on protein thermostability [2] and consequently on functional activity, researchers can uncover fundamental principles governing protein folding, molecular interactions, and evolutionary fitness [3,4]. This approach provides critical insights into the intrinsic trade-offs between stability and function, thereby enabling rational protein engineering and aiding the interpretation of pathogenic or disease associated variants.

The impact of mutations on protein thermostability has been extensively studied across diverse protein systems [5–9]. A protein's ability to remain properly folded and functionally active at elevated temperatures is governed by its thermostability, making it a critical parameter for understanding degradation, aggregation, and denaturation over time. Thermostable proteins are of significant industrial relevance due to their broad-spectrum applications. Their pivotal roles as biocatalysts in chemical synthesis, components in biofuel production, agents in food processing, and additives in detergents are inherently dependent on their structural and functional stability under thermal stress [10]. In the pharmaceutical industry, thermostable therapeutics such as monoclonal antibodies, enzymes, and cytokines exhibit enhanced stability during manufacturing, storage, and transportation, making them more suitable for clinical and commercial applications [11].

Accurate prediction of protein thermostability through computational approaches provides a potentially efficient complement to experimental techniques [12]. In this context, various physics based approaches such as Molecular Mechanics Generalized Born Surface Area (MM/GBSA) [13,14] and Free Energy Perturbation [15–17] methods have gained attention. Unlike data driven machine learning models, which may be constrained by the quality, bias, or representativeness of their training datasets, these methods are founded on fundamental physical principles and provide predictive insights that are independent of empirical training data.

Free energy perturbation (FEP) [18–21] is an alchemical free energy simulation method, firmly rooted in statistical thermodynamics that can be used to estimate free energy differences, provided sufficiently long simulation times, together with enhanced sampling techniques and an accurate potential energy function are employed. It is an explicit solvent, molecular dynamics (MD) based method, offering

a framework for evaluating the thermodynamic impact of amino acid substitutions on protein stability. The term 'alchemical' refers to the generation of a series of nonphysical intermediate states connecting two physical end states typically the wild-type (initial) and mutant (final) forms, facilitating the estimation of the difference between the free energy to fold the wild-type and mutant proteins. The transformation from wild-type to mutant residues, for single mutants(SMs), has been performed in both the folded and unfolded states of the protein that follows a thermodynamic cycle, as illustrated in Fig 1 of the Methodology section. Similarly, the thermodynamic cycle representing double mutations and their corresponding nonadditivity has been depicted in Fig 2 of the Methodology section.

Staphylococcal nuclease (S. nuclease) is a small extracellular protein of 149 amino acid residue length with no cysteines, secreted by *Staphylococcus aureus.* Its ability to degrade both DNA and RNA in the presence of free $Ca^{2+}$ ions, finds various applications in nucleic acid chemistry since 1956 [22,23]. Its simple solution behavior being soluble in both native and denatured states has made it a benchmark system used for studies on protein folding [24,25] and NMR spectroscopy [26]. The experimental values for both single and double mutants (DMs) in S. nuclease were sourced from the extensive work of Shortle et al in S. nuclease [27–29]. A second benchmark protein we focus on in the present study is T4 lysozyme (164 residues) [30], Schellman and co-workers [31] have measured the entropy, enthalpy and free energy change for the wild-type protein and a large number of mutants [32]. A diverse set of experimentally characterized SMs for T4 lysozyme, along with their corresponding $\Delta\Delta G$ values indicating changes in protein stability, has been compiled from the literature [33–41] and considered for the present study.

The present study investigates a total of 62 SMs across two proteins: S. nuclease (38 mutants) and T4 lysozyme (24 mutants), and 45 double mutations of S. nuclease using the Schrödinger (FEP+) and GROMACS packages. These codes are both widely used for studying free energy changes associated with protein-ligand binding in explicit solvent, and more

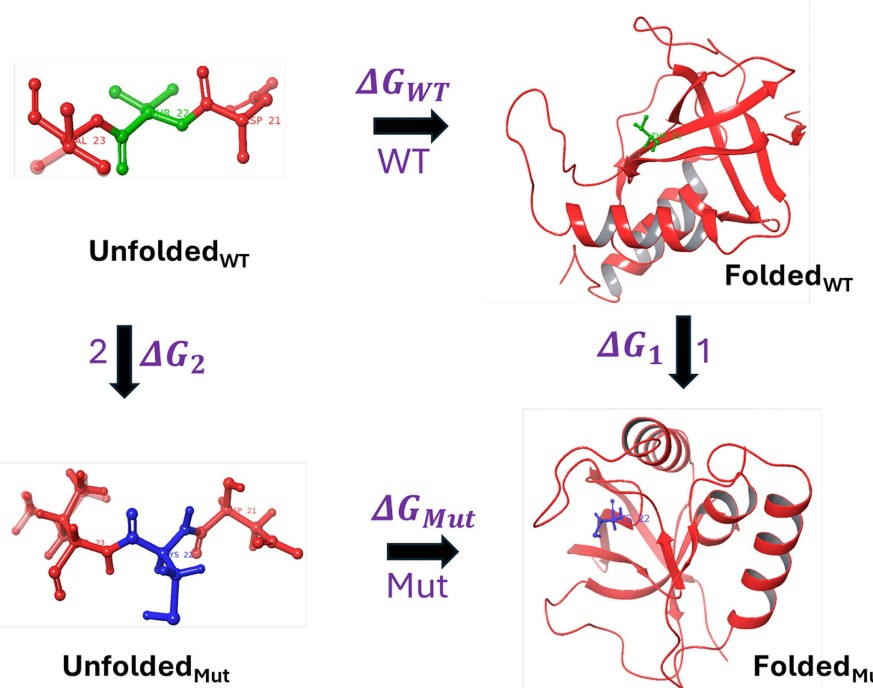

**Fig 1. Thermodynamic Cycle to calculate Stability Change of Single Mutant T22C ($\Delta\Delta G$).** The green color highlights wild-type residue and blue represents the mutated residue, with rest colored as red, being same in both states. The folding free energy change upon single amino acid mutation is $\Delta\Delta G = \Delta G_{Mut} - \Delta G_{WT} = \Delta G_1 - \Delta G_2$.

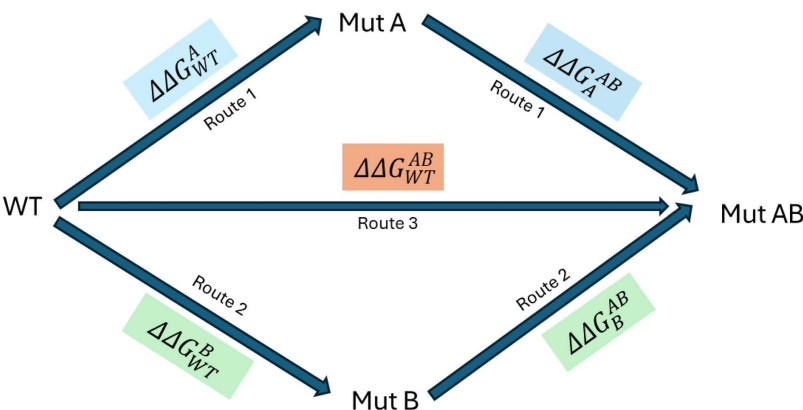

**Fig 2. Double mutant cycle with three different pathways (blue, orange, and green) to calculate the double mutation free energy change and the corresponding nonadditivity ($\delta$).** Here in $\Delta\Delta G_{WT}^{A}$, the Subscript WT = Reference/Initial state, Superscript A = Mutated/Final state and same for others. In principle free energy change from all three routes should be the same. If $\delta_{WT}^{AB} \neq 0$ then mutation A and B are nonadditive, highly correlated and exhibit a thermodynamic coupling.

recently for studying the effects of mutations on protein stability. There have been few efforts to directly compare the performance of these codes on the same benchmark systems.

This study provides the first systematic comparison of free energy change predictions obtained from the Schrödinger and GROMACS simulation platforms across a comprehensive set of single and double mutations in S. Nuclease and a set of single mutations in T4 lysozyme, for which extensive experimental stability data are available. Despite methodological distinctions and differences in the underlying potential energy functions between the two platforms, both yield results that are broadly consistent with experimental measurements and in close agreement with each other.

## Methods

### Dataset selection and preparation

SMs for two proteins under investigation: S. nuclease and T4 lysozyme were curated from the studies by Duan et al (2020) [42] and de Groot et al. (2021) [43]. The initial dataset comprised of 27 SMs of S. nuclease (PDB ID: 1EY0) and 26 SMs of T4 lysozyme (PDB IDs: 2LZM and 1L63), with all experimental measurements performed at pH 7 ± 1. An additional 18 SMs from the dataset associated with PDB: 1STN [43] were incorporated, augmenting the initial 27 mutations and resulting in a total of 45 single mutations analyzed for S. nuclease. After excluding three overlapping mutations, the final dataset comprised of 42 unique SMs. Among these, only 38 mutations were successfully simulated using the Schrödinger and GROMACS platforms, while the remaining four were omitted due to technical issues involving proline residues. Furthermore, a set of 45 DMs [43] for the S. nuclease was also analyzed. These DMs were simulated using only Schrödinger to estimate the free energy changes ($\Delta\Delta G$) and assess their corresponding nonadditive effects. For T4 lysozyme, although the initial dataset contained 26 SMs, two proline-related mutations were excluded due to similar technical constraints, resulting in a final set of 24 single mutations. No double mutant simulations were performed for T4 lysozyme owing to the lack of reliable experimental data for benchmarking. The experimental values were sourced from comprehensive studies by Shortle (for S. nuclease) [27–29] and Matthews (for T4 lysozyme) [34,35,37–41], as mentioned above, serving as benchmark for evaluating the calculated free energy changes from Schrödinger and GROMACS simulations across SM and DM datasets for these two proteins.

Before discussing the methodology for FEP calculations using Schrödinger and GROMACS, it is important to first understand the underlying thermodynamic cycle and the associated coupling parameter that are common to both platforms.

## Description of the alchemical thermodynamic cycles

The alchemical thermodynamic cycles provide a conceptual framework for the transformations from the wild-type to single and double mutants, as illustrated in Fig 1 and Fig 2, respectively. The corresponding nonadditive effects associated with the DMs are also depicted in Fig 2. The transformations include both physical and alchemical (non-physical) states and since free energy is a state function, its value depends only on the initial and final states and hence independent of the pathway taken.

The objective is to compute the folding free energy change resulting from a single mutation, transitioning from the wild-type (WT) to the mutant (Mut) state. Rather than directly simulating the folding process for each state, which is computationally intensive, an alchemical approach has been employed. As illustrated in Fig 1, the physical pathways represent the folding of the WT and Mut state from their respective unfolded structure, with the associated folding free energy denoted as $\Delta G_{WT}$ (path WT) and $\Delta G_{Mut}$ (path Mut) respectively. The alchemical pathways consist of the free energy associated with mutating WT to Mut in the folded state denoted as $\Delta G_1$ (path 1) and in the unfolded state denoted as $\Delta G_2$ (path 2). Since the free energy is a state function, the sum of the $\Delta G$ around the thermodynamic cycle is zero. Consequently, the effect of the mutation on protein stability ($\Delta\Delta G$) can be estimated using Equation 1, which relates the differences in folding free energies of the WT and mutant states via the alchemical transformations.

$$\Delta\Delta G = \Delta G_{Mut} - \Delta G_{WT} = \Delta G_1 - \Delta G_2 \tag{1}$$

The unfolded state of the protein, for both the wild-type and mutant forms, is represented by a capped tripeptide, which preserves the original sequence environment by placing the residue of interest in the central position. The use of a capped tripeptide offers a more realistic and chemically accurate representation for alchemical transformations compared to an uncapped peptide, by mimicking the local backbone environment and minimizing end effects [44].

For DMs an additional thermodynamic cycle has been constructed, as shown in Fig 2. In this cycle, the transformation from the wild-type (WT) to the double mutant (Mut AB) final state can proceed via three distinct routes. In route 1, the transformation proceeds from WT to Mut A, followed by a second mutation to reach state Mut AB. In Route 2, the path goes from WT to Mut B, and then to Mut AB. Route 3 represents a direct transformation from WT to Mut AB, where both mutations A and B are introduced simultaneously. Since free energy is a state function, the total free energy change associated with the double mutation, denoted as ($\Delta\Delta G_{WT}^{AB}$ in Equation 2), must be independent of the transformation path. Therefore, all three routes should theoretically yield identical $\Delta\Delta G_{WT}^{AB}$ values. However, in practice, Route 3 (the direct transformation) is employed because experimentally determined PDB structures for the mutated states A or B are not available

$$\Delta\Delta G_{WT}^{AB} = \Delta\Delta G_{WT}^{A} + \Delta\Delta G_{A}^{AB} = \Delta\Delta G_{WT}^{B} + \Delta\Delta G_{B}^{AB} \tag{2}$$

In the case of double mutations, the combined effect may deviate from the sum of the individual single mutation effects, a phenomenon referred to as nonadditivity ($\delta_{WT}^{AB}$ in Equation 3). When $\delta_{WT}^{AB} \neq 0$, the two mutations are energetically coupled and therefore nonadditive. Conversely, if $\delta_{WT}^{AB} = 0$, the mutations are additive and independent. Within the framework of alchemical FEP, the contribution from the unfolded state is assumed to be additive. Consequently, it cancels out during the calculation of nonadditivity, simplifying the analysis. As a result, the nonadditivity is evaluated solely based on the free energy differences between the single and double mutations in the folded state as formally represented in Equation 3.

$$\delta_{WT}^{AB} = \Delta\Delta G_{WT}^{AB} - (\Delta\Delta G_{WT}^{A} + \Delta\Delta G_{WT}^{B}) \tag{3}$$

**Coupling parameter (λ).** In alchemical free energy simulations, the coupling parameter (λ) is a dimensionless variable used to denote a transition between two thermodynamic states commonly, state A (wild-type) and state B (mutant).[18] As λ progresses from 0 to 1, the system's Hamiltonian is gradually modified in a series of steps from $H_A$ to $H_B$. Mathematically, the interpolated Hamiltonian is represented in Equation 4.

$$H(\lambda) = (1 - \lambda)H_A + \lambda H_B \qquad (4)$$

This parameter governs the pattern of the transformation and serves as a foundational component for computing free energy differences using FEP methods.

With the theoretical framework established, the following sections provide a detailed description of the methodologies employed on each platform. First, we outline the workflow implemented in Schrödinger, followed by the methodology employed in GROMACS for FEP calculations.

### FEP calculations using Schrödinger (FEP+)

The FEP+ module from the Schrödinger Release 2024−3 (FEP+, Schrödinger, LLC, New York, NY, 2024) was employed to perform FEP calculations. Initial protein structures for different proteins were retrieved from the Protein Data Bank (PDB) and prepared using Maestro's Protein Preparation Wizard. The crystal water molecules were retained, hydrogen atoms were added, and hydrogen-bonding networks were optimized. PropKa was used to predict the ionization states of residues at pH 7.0. Subsequently, the structures underwent restrained minimization of heavy atoms using the OPLS4 force field [45], and the process was terminated once the root mean square deviation (RMSD) of heavy atoms reached 0.3 Å. Each system was solvated using a cubic SPC water box (default water model), extending 5 Å from the protein in all directions. To simulate physiological conditions, counter ions $Na^+$ or $Cl^-$ were added to both folded and unfolded systems, followed by the addition of excess $Na^+$ and $Cl^-$ ions to achieve a final salt concentration of 0.15 M NaCl. All ions were randomly distributed throughout the simulation box. For unfolded models, capped tripeptides extracted from the crystal structures of the corresponding native proteins, containing the mutation site at the central position in the peptide were used [42]. The capping groups consisted of an acetyl group at the N-terminus and an N-methyl group at the C-terminus.

The protein FEP+ calculations were performed using a dual topology algorithm, in which multiple MD simulations were performed to transform the system from wild-type (WT) to the mutated state. During this alchemical transformation, the electrostatic and van der Waals interactions of the WT side chain gradually turned off, while those of the mutant side chain were simultaneously turned on. The mutated residue was included in the Replica Exchange Solute Tempering (REST2) region [46,47], which effectively raises the temperature for enhanced sampling of that residue. The FEP+/REST2 framework combines solute tempering with replica exchange to improve sampling of sidechain conformational flexibility near the mutation site, typically within 5 Å. The solvated systems were then relaxed and equilibrated using the default protocol of the Desmond Molecular Dynamics System (D. E. Shaw Research, New York, NY) [48] as implemented in Maestro. This protocol involves a series of energy minimizations and short MD simulations with restraints. Each perturbation was simulated over 16–24 λ windows (where λ denotes the alchemical intermediate states or coupling factor as explained above), depending on the mutation type. The charge conserving mutations were simulated using 16 λ windows while charge altering mutations used 24 λ windows. The λ schedule followed the default settings in Schrödinger FEP+. Electrostatic and van der Waals transformations were integrated across the same λ schedule. Each λ-window was simulated for 10 ns which was extended up to 100 ns to ensure convergence for challenging cases. For charge changing mutations, a co-alchemical water approach was employed to mitigate long range electrostatic artifacts. The electrostatic interactions were handled using the Particle Mesh Ewald (PME) approach [49,50] with a real-space cutoff of 1.0 nm. Van der Waals interactions were also gradually shifted to zero at this same cutoff distance. Final production simulations were conducted with the default ensemble using the Desmond engine and free energy differences between alchemical states were computed using Multi-state Bennett Acceptance Ratio (MBAR) [51] in Schrödinger. Predicted FEP uncertainties were calculated using the cycle closure method [46], as implemented in Schrödinger's Protein FEP+.

### FEP calculations using GROMACS

The alchemical FEP calculations for both protein systems were also performed using GROMACS 2022.6 [52,53] which included 38 SMs from S. nuclease and 23 SMs from T4 lysozyme. The dual topology and the hybrid coordinate files for

the mutated protein were generated using the pmx server [54]. Similarly, the tripeptide structures, extracted from the full length protein using Maestro, were processed by pmx server to obtain their corresponding dual topology representations and hybrid tripeptides. The AMBER99sb*ILDN force field [55–57] has been applied for these transformation in both the cases as this force field has been widely adopted for FEP calculations, as supported by several studies [15,58]. The crystal waters from the protein structure were retained during topology processing of the hybrid protein. Each hybrid system (protein and tripeptide) was solvated in a TIP3P [59] water box with a 10 Å buffer. The counterions Na$^+$ or Cl$^-$ were added to both the systems to neutralize the charge and achieve the final salt concentration of 0.15 M, consistent with standard FEP simulation practices. The retention of crystallographic water and the use of 0.15 M salt concentration align with established FEP protocols in both Schrödinger and GROMACS frameworks.

Then the hybrid protein and tripeptide systems underwent energy minimization using the steepest descent algorithm across 27 λ windows (from λ = 0 to λ = 26). The λ schedule varied as 0.00 0.0125 0.025 0.0375 0.05 0.10 0.15 0.20 0.25 0.30 0.35 0.40 0.45 0.50 0.55 0.60 0.65 0.70 0.75 0.80 0.85 0.90 0.95 0.9625 0.975 0.9875 1.00 from 0 to 26 λ windows correspondingly. The 27 λ windows were partitioned into electrostatic decoupling (10 λ windows) and van der Waals decoupling (17 λ windows), with soft-core potentials applied to enable smooth interpolation of non-bonded interactions. This was followed by an additional minimization using the l-bfgs integrator for further relaxation. Subsequently, the systems were equilibrated under the NVT ensemble for 10 ps using the velocity-rescaling thermostat [60]. This step was followed by NPT equilibration ensemble, performed sequentially using the Berendsen and Parrinello-Rahman barostats [61] each for 30 ps. The electrostatic interactions were treated using the Particle Mesh Ewald (PME) method [49,50] and a real space cutoff of 1.0 nm was applied. The van der Waals interactions were gradually shifted to zero at the same cutoff distance (1.0 nm). All bonds involving hydrogen atoms were constrained using the LINCS [50] algorithm (order = 6). Following equilibration, replicas were simulated for 50 ns with NPT ensemble for both the hybrid protein and the tripeptide systems. The free energy differences between the wild-type and the mutant forms were computed using the Bennett Acceptance Ratio (BAR) [62,63] method, implemented via the gmx_bar module in GROMACS. Alchemical free energy calculations were also analyzed using the pymbar package based on $\partial H/\partial\lambda$ data obtained from GROMACS simulations. The robustness of the free energy estimates by MBAR was verified through two primary diagnostics: 1) To ensure reliable reweighting of configurations across states, $\partial H/\partial\lambda$ overlap matrices were generated for representative mutations. These matrices (S5 and S6 Figs) confirm appreciable overlap between adjacent λ-windows for both the S. nuclease (e.g., T41C, T41V) and T4 lysozyme (e.g., T59G, G77A) systems. 2) Convergence was assessed by monitoring the change of ΔG as a function of simulation time. Representative convergence plots (S7 and S8 Figs) for mutants such as L25I and T33V (S. nuclease) and S44A and D47A (T4 lysozyme) demonstrate that the free energy differences reach a stable plateau. This behavior indicates that convergence was achieved over the simulation time for both the tripeptide reference and the full protein. Predicted uncertainties were estimated by partitioning each simulation trajectory into three equal segments and calculating the standard deviation of the ΔΔG values across these sub-blocks.

In summary, to assess the impact of mutations on protein free energy landscapes, we performed a comparative analysis of single and double mutations FEP protocols implemented in GROMACS and Schrödinger's FEP+ framework. Although both platforms utilize alchemical transformation methods, they differ markedly in system setup, topology handling, and sampling strategies. This study aims to evaluate the consistency and reliability of predicted $\Delta\Delta G$ values across these two distinct, widely used, computational platforms.

## Results and discussion

Experimental determination of mutation-induced changes in protein thermostability, typically using chemical or thermal denaturation techniques, remains the gold standard for validation. However, these methods are often time-consuming, resource-intensive, and low-throughput, making them more difficult for large-scale mutational screening. Moreover, while such methods yield valuable thermodynamic data, they offer limited insight into the atomic level interactions underlying protein stability changes. In contrast, computational methods such as FEP, which leverage all-atom MD simulations,

potentially provide a faster, scalable, and cost-effective alternative provided they have sufficient accuracy. These methods not only enable high-throughput *in silico* mutagenesis but also deliver atomistic insights into stability mechanisms, allowing simulations of various protein conformational states to better understand folding landscapes and thermostability. Recent advances in high performance computing including GPU acceleration as well as improvements in force fields and simulation algorithms, have enhanced the predictive accuracy of these computational approaches.

In this work, we employed two widely used MD platforms: Schrödinger and GROMACS, for FEP simulations, each offering distinct advantages and limitations. GROMACS is an open-source, freely available MD engine noted for its flexibility, extensive customizability, and support for a broad range of force fields including AMBER, CHARMM, OPLS and GROMOS. However, its implementation for FEP workflows requires manual setup, command-line proficiency, and greater time investment, making it less accessible to non-expert users. In contrast, Schrödinger FEP+is a commercial, closed-source platform that supports only the modified OPLS force field. While it offers less flexibility, its user-friendly graphical interface, built-in FEP workflow as FEP+ module, and enhanced sampling methods such as REST (Replica Exchange with Solute Tempering) streamline the simulation process and significantly reduce setup complexity and computational overhead. Despite the widespread use of both platforms for FEP-based protein stability predictions, a systematic comparison of their performance using experimental data as a benchmark has been lacking. In this work, we address this gap by directly evaluating and comparing free energy changes for single and double mutations using Schrödinger and GROMACS against experimental thermostability data, thereby providing insights into their relative accuracy, usability, and computational efficiency.

The free energy changes ($\Delta\Delta G$) for 38 single mutations of S. nuclease were predicted using the two MD platforms: Schrödinger and GROMACS. These results, along with the $\Delta\Delta G$ values calculated via MBAR from the GROMACS data, are provided in Supporting Information (S1 Table). Correlation plots comparing experimentally determined free energy changes with predictions calculated using Schrödinger and GROMACS are shown in Fig 3A and 3B respectively. Both platforms demonstrated state-of-the-art accuracy in reproducing experimental measurements, with Pearson correlation coefficients of 0.86 for Schrödinger (Fig 3A) and 0.87 for GROMACS (Fig 3B) between calculated and experimental $\Delta\Delta G$ values, indicating strong linear agreement. Additionally, the mean absolute difference also referred to as the average unsigned error (AUE), was further calculated to evaluate predictive accuracy. The AUE values were found to be 0.99 kcal/mol for Schrödinger and 0.61 kcal/mol for GROMACS, both within the generally accepted accuracy threshold of ~1 kcal/mol for reliable free energy predictions. Notably, these AUE values are comparable to those reported for FEP+ protocol in predicting small molecule protein relative binding affinities [64,65], supporting the robustness of our results. Benchmarking results revealed strong agreement between computational predictions and experimental values. Specifically, the RMSE was 0.13 kcal/mol for GROMACS (via BAR) and 0.24 kcal/mol for Schrödinger. Additionally, a secondary check using MBAR on the GROMACS data resulted in an RMSE of 0.13 kcal/mol. Correlation was further evaluated using Kendall's tau, which yielded values of 0.54 (GROMACS/BAR), 0.62 (Schrödinger), and 0.50 (GROMACS/MBAR). These statistical indicators, provided in S1 Table, highlight the high predictive accuracy of both platforms. The strong Pearson r of 0.87 between the previously reported [42] and our calculated folding free energy changes ($\Delta\Delta G$) highlights the reliability and robustness of these calculations, as shown in S1 Fig and S1 Table. The high correlation and low AUE observed in this study are consistent with previously reported FEP studies [42,66], where correlations between computed and experimental $\Delta\Delta G$ values ranged from 0.64 to 0.82, further validating the reliability of both platforms. Overall, these findings establish both Schrödinger and GROMACS as accurate and reliable tools for predicting protein thermostability changes induced by single mutations using FEP-based approaches.

The analysis was extended to evaluate folding free energy change for 45 DMs of S. nuclease represented as $\Delta\Delta G_{WT}^{AB}$. The values calculated using Schrödinger and previously reported values using GROMACS [43] alongside experimental measurements, which continue to serve as the benchmark, were summarized in S2 Table. The calculated $\Delta\Delta G_{WT}^{AB}$ using Schrödinger yielded a Pearson correlation coefficient of 0.74 when compared with the experimental

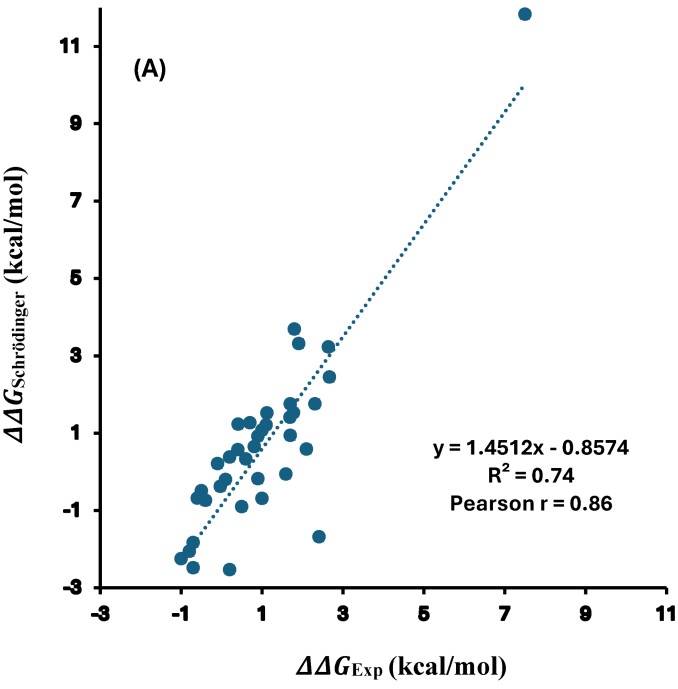

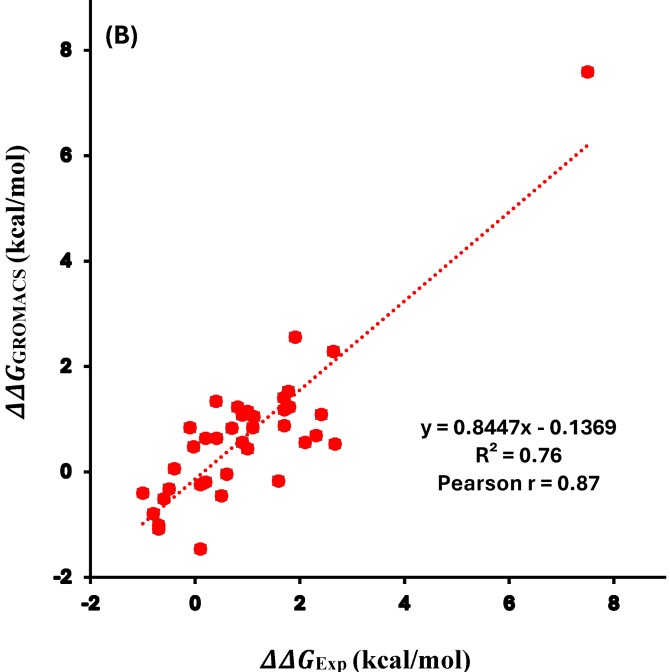

**Fig 3. Correlation plots of the free energy changes for 38 SMs for S. nuclease between the experimental values and calculated ones from (A) Schrödinger and (B) GROMACS.** Pearson correlation coefficient (r) and coefficient of determination ($R^2$) are indicated. The data plotted here is provided in S1 Table.

data (Fig 4A). A similarly strong correlation (r = 0.71) was observed between the previously reported GROMACS values and experimental results, as shown in Fig 4B. The AUE was found to be 0.79 kcal/mol for Schrödinger and 0.76 kcal/mol for GROMACS, both within the generally accepted threshold of 1 kcal/mol. Moreover, we compared previously reported GROMACS values with those calculated using Schrödinger, yielding a Pearson correlation of 0.71 (S2 Fig). Comparable accuracy and low AUE values demonstrate that both Schrödinger (OPLS4 force field) and GROMACS (AMBER99sb*ILDN force field) gave consistent and reliable results for double mutations in the S. nuclease protein. The results demonstrate high predictive accuracy, reflected in low RMSE values of 0.13 kcal/mol for both GROMACS and Schrödinger when compared to experimental data. This strong agreement is further supported by Kendall's tau (τ) values of 0.52 for GROMACS and 0.57 for Schrödinger. These metrics, provided in S2 Table, validate the reliability of both software platforms for benchmarking free energy calculations.

In the case of DMs, the free energy changes can deviate significantly from the sum of the individual single mutations, a phenomenon known as nonadditivity ($\delta_{WT}^{AB}$). The corresponding nonadditivities for the 45 DMs have been calculated employing the FEP+ protocol in Schrödinger. For comparison, previously reported free energy changes calculated using GROMACS [43] were included. Experimental values continue to serve as the benchmark for this study, consistent with the analyses of single and double mutations. Schrödinger predictions of nonadditivity yielded a Pearson r of 0.79 with experimental data (Fig 5A), whereas the previously reported GROMACS values showed a correlation of 0.63 (Fig 5B). The AUE was found to be 0.31 kcal/mol for Schrödinger and 0.35 kcal/mol for GROMACS relative to experimental values. Additionally, a Pearson r of 0.61 was observed for the nonadditivity values ($\delta_{WT}^{AB}$) of 45 DMs of S. nuclease between previously reported GROMACS predictions and our calculated Schrödinger values, as illustrated in S3 Fig. All these nonadditivity values obtained from experimental measurements, GROMACS and Schrödinger calculations were tabulated in S3 Table. The calculated RMSE values were notably low, at 0.07 kcal/mol for GROMACS and 0.06 kcal/mol for Schrödinger relative to experimental data, indicating high predictive accuracy. As an additional metric, Kendall's tau (τ) was found to be moderate, 0.48 for GROMACS and 0.44 for Schrödinger. These results, which support the benchmarking of both free energy calculation platforms, are summarized in S3 Table.

We have extended our analysis further to another widely studied protein system, T4 lysozyme in order to gain more confidence in thermostability predictions using computational approaches. The dataset consists of 24 single mutations for T4 lysozyme protein, for which free energy changes ($\Delta\Delta G$) were calculated using both the FEP+ module of Schrödinger and the GROMACS. However, DMs were not included for the T4 lysozyme system, as experimental $\Delta\Delta G_{WT}^{AB}$ values were not available.

To assess the agreement between calculated and experimental values, correlation analyses were performed. Fig 6A shows the correlation between experimental $\Delta\Delta G$ values and those calculated using Schrödinger (Pearson r = 0.85), while Fig 6B presents the correlation between experimental values and GROMACS predictions (Pearson r = 0.80). The corresponding AUEs were 0.63 kcal/mol for Schrödinger and 0.55 kcal/mol for GROMACS. A direct comparison of computed $\Delta\Delta G$ values with Schrödinger and GROMACS yielded a Pearson r of 0.81 (Fig 6C). Furthermore, a Pearson r of 0.96 was observed between previously reported[42] Schrödinger results and the $\Delta\Delta G$ values calculated using Schrödinger in this study (S4 Fig). All free energy changes: experimental, calculated via GROMACS and Schrödinger, and previously reported Schrödinger values were summarized in S4 Table. Collectively, these results demonstrate the effectiveness in modeling free energy changes arising from single mutations in T4 lysozyme [42] by GROMACS and Schrödinger. Additional performance metrics, including RMSE and Kendall's tau, are provided in the sub-tables of S4 Table. The low RMSE values-0.13 kcal/mol for GROMACS (BAR), 0.16 kcal/mol for Schrödinger, and 0.13 kcal/mol for GROMACS (MBAR) demonstrate strong agreement with experimental data and high predictive accuracy. Similarly, Kendall's tau values of 0.61, 0.68, and 0.61, respectively, further validate the benchmarking of these two software platforms for free energy calculations.

To investigate potential correlations, we analyzed prediction errors as a function of mutation type and location (S5-S6 Tables). Prediction accuracy was highest for mutations involving no change in side-chain size, with errors increasing

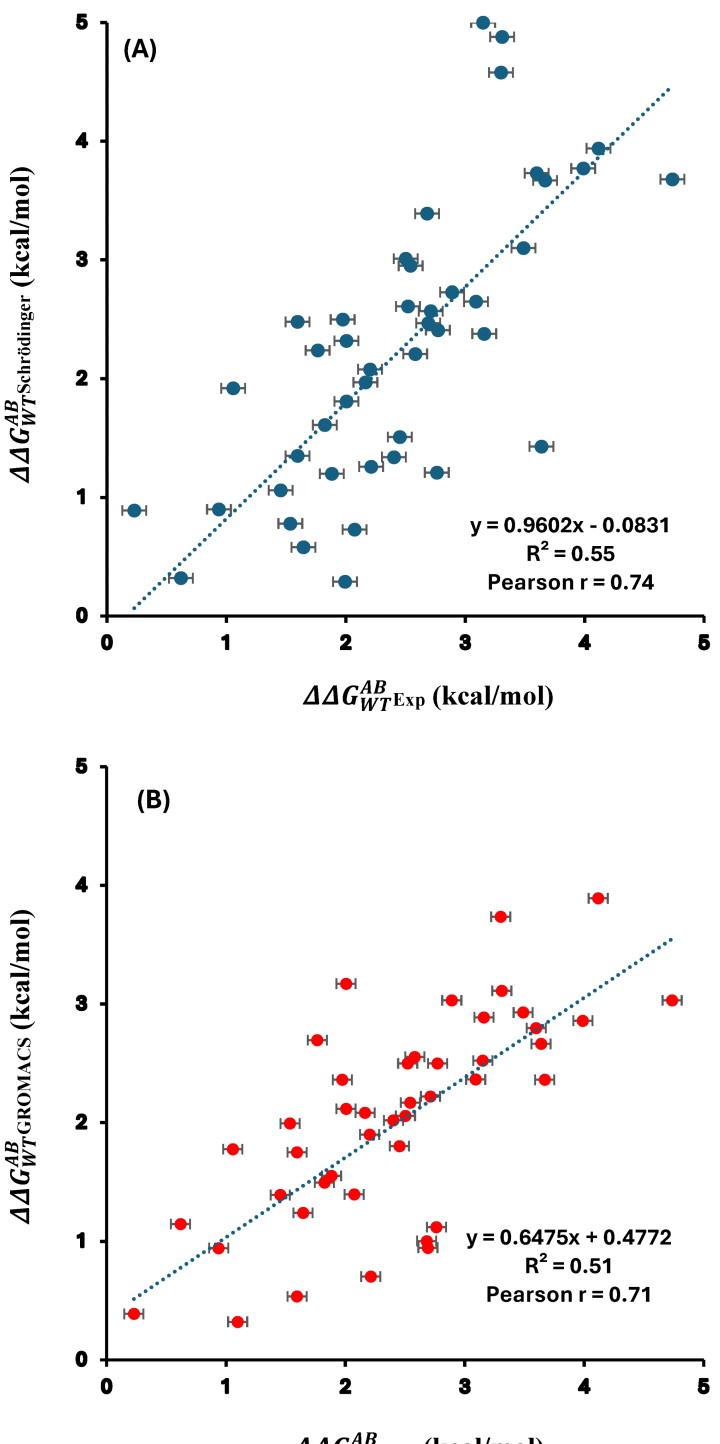

**Fig 4. Correlation plots of the free energy changes for S. nuclease's 45 DMs between the experimental values and calculated ones with (A) Schrödinger and (B) GROMACS.** Pearson correlation coefficient (r) and coefficient of determination (R²) are indicated. The data plotted here is provided in S2 Table.

**Fig 5. Correlation plots for the corresponding nonadditivity for S. nuclease's 45 DMs between experimental values and calculated ones with (A) Schrödinger and (B) GROMACS.** Pearson correlation coefficient (r) and coefficient of determination ($R^2$) are indicated. The data plotted here is provided in S3 Table.

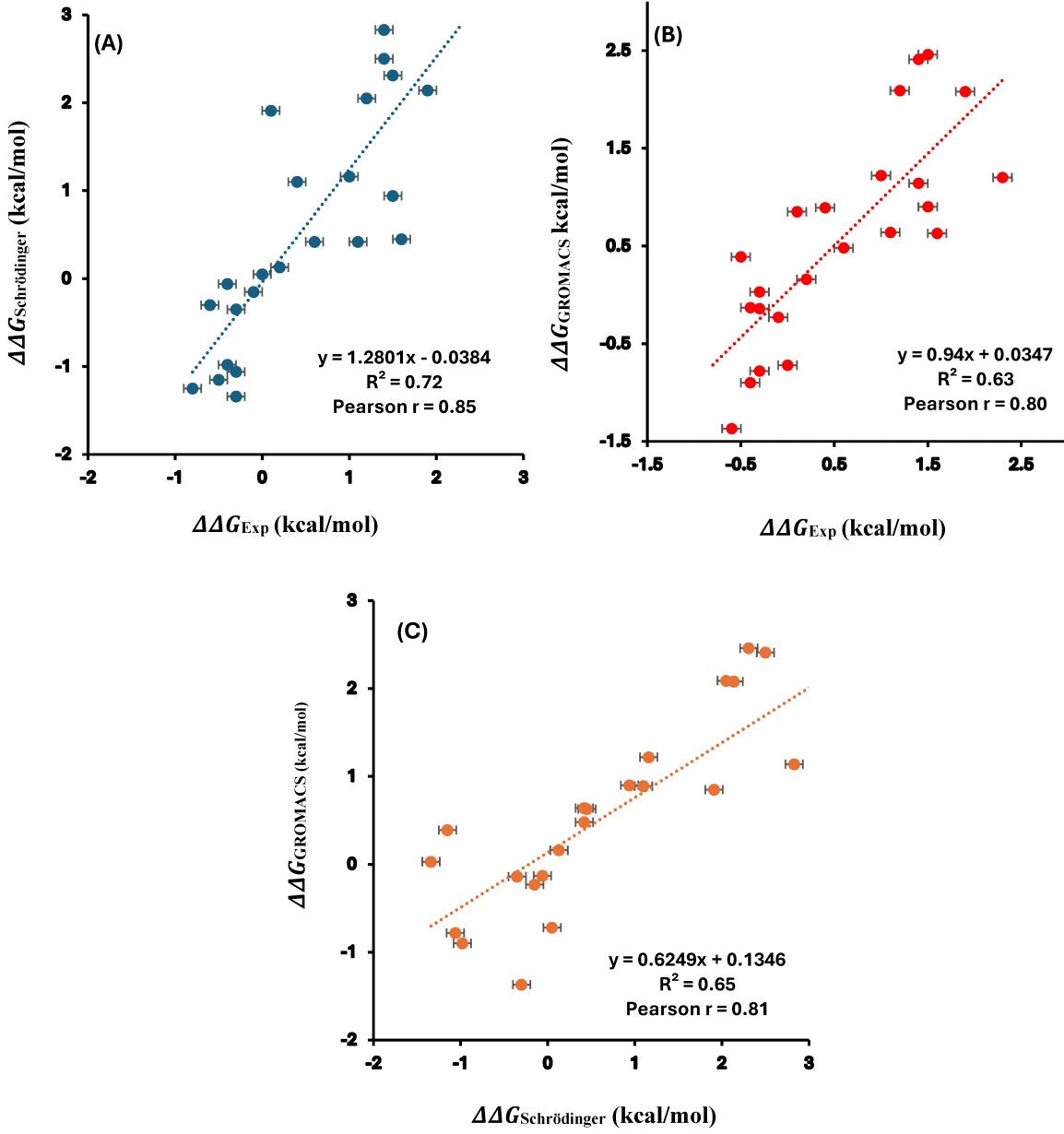

**Fig 6. Correlation plots of the free energy changes for T4 lysozyme 24 SMs between the experimental and calculated values with (A) Schrödinger and (B) GROMACS and (C) Schrödinger vs GROMACS.** Pearson correlation coefficient (r) and coefficient of determination (R²) are indicated. The data plotted here is provided in S4 Table.

significantly as side-chain bulk increased. Additionally, charged mutations exhibited larger errors than neutral ones. Location-wise, surface mutations generally yielded higher errors than buried residues.

## Conclusion

FEP methods are increasingly employed to estimate changes in protein thermostability caused by mutations. In this study, we compared two widely used MD platforms: GROMACS and Schrödinger's FEP+, to perform FEP-based

calculations for SMs of S. nuclease and T4 lysozyme proteins. Additionally, FEP-based predictions were extended to DMs of S. nuclease, including the assessment of their corresponding nonadditivity effects. Our results showed strong correlation between experimental and computed free energy changes $(\triangle\triangle G)$ for 62 SMs in two different proteins using GROMACS and Schrödinger. Schrödinger accurately captured the free energy changes $(\triangle\triangle G_{WT}^{AB})$ and their corresponding nonadditivities $(\delta_{WT}^{AB})$ for 45 DMs of S. nuclease protein, showing good correlation with experimental values. A strong Pearson correlation for free energy changes and nonadditivity for 45 DMs of S. nuclease, has also been observed between the previously reported GROMACS values and those calculated in this study using Schrödinger. The findings demonstrate the robustness of both Schrödinger and GROMACS in modeling single and double mutations, accurately predicting their free energy changes and nonadditivity. The strong correlation between experimental and computed free energy changes further highlights the consistency and accuracy of these platforms in capturing the thermodynamic effects of mutations.

The choice between two platforms depends on factors such as the complexity of the system under study, the cost of the software involved, preference towards a specific force field or enhanced sampling technique, and user familiarity. While FEP is not a substitute for experimental validation, it offers a rapid, scalable, and mechanistically informative approach to mutation analysis in proteins. Importantly, FEP enables atomic level insights into the effects of mutations, which are not directly accessible through experimental denaturation assays alone. Therefore, integrating FEP-based computational predictions (either by GROMACS or Schrödinger) with experimental measurements presents a powerful and complementary strategy for protein engineering and thermostability optimization.

## Supporting information

**S1 Table. Folding free energy changes** $(\triangle\triangle G)$ **in kcal/mol for 38 SMs of the S. nuclease protein.** The experimental values and previously reported values using Schrödinger or GROMACS alongside values calculated in this study using both platforms are provided for comparison.
(PDF)

**S1 Fig. Correlation plot of free energy changes** $(\triangle\triangle G)$ **for 38 SMs in the S. nuclease protein, comparing previously reported values using Schrödinger or GROMACS with values calculated in this study using Schrödinger.** Pearson correlation coefficient (r) and coefficient of determination (R²) are indicated.
(PDF)

**S2 Table. Folding free energy changes** $(\triangle\triangle G_{WT}^{AB})$ **in kcal/mol for 45 DMs of the S. nuclease protein.** The experimental values and previously reported values using GROMACS alongside values calculated in this study using Schrödinger are shown for comparison.
(PDF)

**S2 Fig. Correlation plot of free energy changes** $(\triangle\triangle G_{WT}^{AB})$ **for 45 DMs in the S. nuclease protein, comparing previously reported values from GROMACS with values calculated in this study using Schrödinger.** Pearson correlation coefficient (r) and coefficient of determination (R²) are shown.
(PDF)

**S3 Table. Nonadditivity values** $(\delta_{WT}^{AB})$ **in kcal/mol for 45 DMs of the S. nuclease protein.** The experimental data and previously reported GROMACS values are presented alongside values calculated in this study using Schrödinger for comparative analysis.
(PDF)

**S3 Fig. Correlation plot of nonadditivity ($\delta_{WT}^{AB}$) for 45 DMs in the S. nuclease protein, comparing previously reported values from GROMACS with values calculated in this study using Schrödinger.** Pearson correlation coefficient (r) and coefficient of determination ($R^2$) are shown.
(PDF)

**S4 Table. Folding free energy changes ($\triangle\triangle G$) in kcal/mol for 24 SMs of the T4 lysozyme protein.** The experimental values and previously reported values using Schrödinger alongside values calculated in this study using GROMACS and Schrödinger are provided for comparison.
(PDF)

**S4 Fig. Correlation plot of free energy changes ($\triangle\triangle G$) for 24 SMs in the T4 lysozyme protein, comparing previously reported values with calculated values from Schrödinger.** Pearson correlation coefficient (r) and coefficient of determination ($R^2$) are shown.
(PDF)

**S5 Fig. Overlap matrix plots for representative *S. nuclease* mutations (T41C and T41V) with all first off-diagonal entries well above 0.03, the suggested threshold.** Each of the four Figs consists of two panels: the upper panel displays the overlap for the tripeptide, while the lower panel displays the overlap for the full protein system.
(PDF)

**S6 Fig. Overlap matrix plots for representative T4 lysozyme mutants T59G and G77A with all first off-diagonal entries well above 0.03, the suggested threshold.** Each of the four Figs consists of two panels: the upper panel displays the overlap for the tripeptide, while the lower panel displays the overlap for the full protein system.
(PDF)

**S7 Fig. Free energy convergence for representative S. nuclease mutants L25I and T33V.** The plots show ΔΔG versus simulation time across four Figs. In each case, the top panel corresponds to the tripeptide simulation (unfolded state proxy) and the bottom panel corresponds to the full protein simulation (folded state).
(PDF)

**S8 Fig. Free energy convergence for representative T4 lysozyme mutants S44A and D47A.** The plots show ΔΔG versus simulation time across four Figs. In each case, the top panel corresponds to the tripeptide simulation (unfolded state proxy) and the bottom panel corresponds to the full protein simulation (folded state).
(PDF)

**S5 Table. Calculated free energy changes (kcal/mol) for 38SMs of S. nuclease using Schrödinger and GROMACS, compared against experimental values.** The mutations are categorized by charge, size, and location within the protein structure.
(PDF)

**S6 Table. Calculated free energy changes (kcal/mol) for 24 SMs of T4 lysozyme using Schrödinger and GROMACS, compared against experimental values.** The mutations are categorized by charge, size, and location within the protein structure.
(PDF)

## Author contributions

**Conceptualization:** Qinfang Sun, Ronald M. Levy.

**Data curation:** Shivani Gupta.

**Formal analysis:** Shivani Gupta, Qinfang Sun.

**Funding acquisition:** Ronald M. Levy.

**Investigation:** Shivani Gupta, Qinfang Sun, Ronald M. Levy.

**Methodology:** Shivani Gupta, Qinfang Sun.

**Project administration:** Ronald M. Levy.

**Resources:** Qinfang Sun.

**Software:** Shivani Gupta, Qinfang Sun.

**Supervision:** Qinfang Sun, Ronald M. Levy.

**Validation:** Shivani Gupta, Qinfang Sun, Ronald M. Levy.

**Visualization:** Shivani Gupta, Qinfang Sun, Ronald M. Levy.

**Writing – original draft:** Shivani Gupta.

**Writing – review & editing:** Shivani Gupta, Qinfang Sun, Ronald M. Levy.

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
