## [Decision Letter · Decision Letter 0]

11 Nov 2025

Dear Dr. Levy,

Thank you for submitting your manuscript to PLOS ONE. After careful consideration, we feel that it has merit but does not fully meet PLOS ONE’s publication criteria as it currently stands. Therefore, we invite you to submit a revised version of the manuscript that addresses the points raised during the review process.

We look forward to receiving your revised manuscript.

Kind regards,

Yong Wang

Academic Editor

PLOS ONE

Journal Requirements:

“NIH Grant R35 GM132090”

“This work was supported in part by NIH Grant R35 GM132090.”

“NIH Grant R35 GM132090”

5. Please provide a complete Data Availability Statement in the submission form, ensuring you include all necessary access information or a reason for why you are unable to make your data freely accessible. If your research concerns only data provided within your submission, please write "All data are in the manuscript and/or supporting information files" as your Data Availability Statement.

Reviewer's Responses to Questions

**Comments to the Author**

1. Is the manuscript technically sound, and do the data support the conclusions?

Reviewer #1: Partly

Reviewer #2: Yes

2. Has the statistical analysis been performed appropriately and rigorously?

Reviewer #1: No

Reviewer #2: Yes

3. Have the authors made all data underlying the findings in their manuscript fully available?

Reviewer #1: No

Reviewer #2: Yes

4. Is the manuscript presented in an intelligible fashion and written in standard English?

Reviewer #1: Yes

Reviewer #2: Yes

Reviewer #1: The paper is about the benchmark of mutation ddG on two software platforms (FEP+ and Gromacs), on 2 protein system.

1. Correlation should not be the main matrix of comparison. RMSE, kendall tau, or AUE (already being used in the results and discussion), can also be used to evaluate the quality of free energy calculations. And Correlation of ddG is not well defined, because of the following reason. There is not really a direction of A to B and B to A, but plotting some ddG edges of A to B or B to A will give different correlation analysis results.

2. In the method section of “FEP calculations using GROMACS”. Please more clearly explain: 1. What soft-core potential is used? Beutler or Gapsys 2. Soft-core potential is applied to both vdw and coulomb or vdw only. 3. “from λ= 0 to λ = 26 …” is confusing and inconsistent with equation 4 in which λ ϵ [0,1]. 4. Are those 27 windows simulated separately or together with Hamiltonian replica exchange. Hamiltonian replica exchange is an available enhanced sampling method in the native Gromacs mdrun. It does not affect simulation through put while bringing in sampling enhancement. If not using, please justify the reason for not using. 5. “AMBER99sb*ILDN” was stated in method, but “Amber99SB-ILDN” was stated in the abstract. 6. How was the charge neutralized when a charge mutation was applied? 7. Which ion parameters were used? 8. Please clarify the exact lambda schedule for example: 0.0, 0.05, … 1.00 for electrostatic … 9. As a benchmark, all the input files (structure, Gromacs mdp files for every steps), analysis files, and results files (e.i. ddG values in csv file) should be provided, either in SI or a openly available repository. Not only should the value of free energy estimation be reported, the uncertainties of each free energy estimation should also be reported.

3. Please make the x-y axis aspect equal when comparing experiment and computed delta G. As 1 kcal/mol in the experiment means the same as 1 kcal/mol in the experiment.

4. Two platforms (FEP+ and gromacs) are using different force fields, enhanced sampling algorithms, and sampling time. From the benchmark results, can we get some insight about the advantages and disadvantages about OPLS4 vs amber99*ildn and REST2 with 16/24 lambda vs 27 lambda with no replica exchange?

5. Does the magnitude of the error correlate with the type of mutation (sides of the amino acid, neutral or charged mutation). Does the error correlate with the place of the mutation (the surface of the protein or in the core of the protein.) With this benchmark, can we get some insight about what kind of mutation is more difficult to calculate with alchemical free energy calculations.

Reviewer #2: This manuscript presents a systematic benchmarking study comparing FEP calculations using Schrödinger FEP+ and GROMACS platforms for predicting mutation-induced stability changes in S. nuclease and T4 lysozyme. The work is methodologically sound and addresses an important gap in the field by directly comparing two widely-used computational platforms. However, several areas require clarification and improvement before publication.

(1)The manuscript reports good correlations between calculated and experimental ΔΔG values. However, I notice that no statistical uncertainties (e.g., standard deviations or confidence intervals) are provided for the computed free energy changes. Given the stochastic nature of molecular dynamics simulations, it would be valuable to assess the statistical reliability of these predictions. Could the authors clarify whether multiple independent simulations were performed for each mutation? If not, it would be feasible to estimate statistical uncertainties using block averaging or bootstrap methods.

(2)The study employs different numbers of λ-windows between the two platforms: 16-24 for Schrödinger and 27 for GROMACS. I am curious about the specific λ-scheduling strategies used in each platform. Could the authors provide details on: (1) the exact distribution of λ values (uniform vs. non-uniform spacing), (2) whether additional λ-windows were placed in critical regions where electrostatic-to-van der Waals transitions occur, and (3) how the different λ-schedules might affect the precision of free energy integration, particularly in regions of poor phase space overlap? Including supplementary data showing dH/dλ distributions or overlap matrices would help readers assess the adequacy of sampling at each λ-window.

(3)I note that Schrödinger (using MBAR) and GROMACS (using BAR) utilized different free energy estimators. While MBAR is theoretically advantageous, its performance often relies on sufficient λ-window sampling. Did the authors evaluate the specific impact of this methodological difference on the comparative results? For instance, would post-processing the GROMACS data with MBAR yield results significantly different from those obtained with BAR?

(4)I notice that the simulation setup differs between platforms regarding the solvent box size: a 5 Å buffer for Schrödinger versus a 10 Å buffer for GROMACS. The authors need to clarify the rationale behind this difference.

(5)The conclusions of the manuscript heavily rely on the convergence of the simulations. Beyond simulation length, the most compelling evidence would be the time-evolution of the free energy for key calculations. The authors need provide convergence plots of ΔΔG vs. simulation time for some representative mutants (e.g., the best and worst predicted cases) to visually demonstrate the convergence and stability of the calculations?

(6) The dataset includes charge-changing mutations, which introduce net charge artifacts under periodic boundary conditions with PME electrostatics—a well-documented source of systematic error in alchemical free energy calculations. I note that the manuscript briefly mentions the use of a "co-alchemical water" approach in Schrödinger to mitigate this issue, but provides no details on its implementation or validation. More critically, there is no discussion of how GROMACS handles net charge corrections for the same mutations. Could the authors clarify: (1) the exact protocol for the co-alchemical water method and whether it fully compensates for finite-size artifacts, (2) whether GROMACS employs any net charge correction schemes (e.g., the analytical correction by Rocklin et al., J. Chem. Theory Comput. 2013, 9, 3072-3083) or simply relies on a neutralizing background plasma, and (3) how these potentially different treatments between platforms might systematically bias the comparative results? This concern is particularly important given the different box sizes noted in point (4), as finite-size effects scale with system size. I would recommend either: (a) a sensitivity analysis quantifying the magnitude of net charge artifacts for a subset of charge-changing mutations, or (b) a thorough literature-based justification demonstrating that the chosen approaches yield negligible systematic errors under the simulation conditions employed.

(7) A fundamental methodological concern is that this study simultaneously compares two different software platforms, two different force fields (OPLS4 in Schrödinger vs. AMBER99SB*ILDN in GROMACS), two different water models (SPC vs. TIP3P), and different enhanced sampling strategies (REST2 in Schrödinger vs. standard MD in GROMACS). This multifactorial design makes it challenging to disentangle whether the observed differences in performance arise from: (1) software implementation details, (2) force field parameterization, (3) water model effects, or (4) sampling methodology. While I recognize that systematically testing all possible combinations may be beyond the scope of this benchmarking study, this confounding factor represents a significant limitation that should be explicitly acknowledged in both the Discussion and Limitations sections.

.

Reviewer #1: **Yes:**Chenggong HuiChenggong HuiChenggong HuiChenggong Hui

Reviewer #2: No

---

## [Author Response · Author response to Decision Letter 1]

12 Feb 2026

Authors: We thank both reviewers for the time and effort spent on reviewing our manuscript. Their comments/questions have been very helpful for improving the paper.

Reviewer #1: The paper is about the benchmark of mutation ddG on two software platforms (FEP+ and Gromacs), on 2 protein systems.

1. Correlation should not be the main matrix of comparison. RMSE, kendall tau, or AUE (already being used in the results and discussion), can also be used to evaluate the quality of free energy calculations. And Correlation of ddG is not well defined, because of the following reason. There is not really a direction of A to B and B to A, but plotting some ddG edges of A to B or B to A will give different correlation analysis results.

Authors: We agree that correlation alone does not fully capture the quality of ΔΔG predictions. In response, we have added RMSE, Kendall’s τ, and AUE as additional metrics to assess the accuracy and consistency of the free energy calculations across platforms on sections to provide a more comprehensive and direction-independent evaluation of performance. We have calculated RMSE and Kendall tau values for the comparisons of the experiments with the results of the simulations; they are listed in table S1-S4 and have also been added on page 9 (line # 314-320), page 10 (line # 341-347), page 10 (line # 361- 366) and page 11 (line # 384-391). All these lines have been highlighted in yellow in the main manuscript. The correlation of ΔΔG (Wild type → Mutated) quantifies the agreement between computational predictions and experimental measurements of Gibbs free energy changes upon mutation.

2. In the method section of “FEP calculations using GROMACS”. Please more clearly explain: 1. What soft-core potential is used? Beutler or Gapsys 2. Soft-core potential is applied to both vdw and coulomb or vdw only. 3. “from λ= 0 to λ = 26 …” is confusing and inconsistent with equation 4 in which λ ϵ [0,1]. 4. Are those 27 windows simulated separately or together with Hamiltonian replica exchange. Hamiltonian replica exchange is an available enhanced sampling method in the native Gromacs mdrun. It does not affect simulation through put while bringing in sampling enhancement. If not using, please justify the reason for not using. 5. “AMBER99sb*ILDN” was stated in method, but “Amber99SB-ILDN” was stated in the abstract. 6. How was the charge neutralized when a charge mutation was applied? 7. Which ion parameters were used? 8. Please clarify the exact lambda schedule for example: 0.0, 0.05, … 1.00 for electrostatic … 9. As a benchmark, all the input files (structure, Gromacs mdp files for every steps), analysis files, and results files (e.i. ddG values in csv file) should be provided, either in SI or a openly available repository. Not only should the value of free energy estimation be reported, the uncertainties of each free energy estimation should also be reported.

Authors: The Methods section has been revised to clarify all points as follows:

Soft-core potential: The Beutler soft-core potential (Beutler et al., J. Comput. Chem. 1994; https://pubs.acs.org/doi/10.1021/ct300220p) is used in GROMACS.

Application: The soft-core potential was applied to both van der Waals and Coulomb interactions to ensure a smooth decoupling process.

λ notation: We have clarified the text to read “from λ index 0 to λ index 26”, corresponding to 27 windows over λ ∈ [0,1].

Not Replica exchange: The 27 windows were simulated separately.

Force field naming: The inconsistency has been corrected to “AMBER99sb*ILDN” throughout the manuscript.

Charge neutralization: For charge-changing mutations, a counter-ion (Na⁺ or Cl⁻) was mutated to water to maintain charge neutrality.

Ion parameters: Na⁺ and Cl⁻ ion parameters compatible with the TIP3P water model were used, consistent with the AMBER99sb*ILDN force field.

λ schedule: The λ schedule has been added explicitly in the revised Methods section. This has been added for Schrodinger and GROMACS on page 6 and 7 in line no. 218-219, and pg- 244-246 respectively.

Data availability: All input files (system structures, GROMACS input and output files, analysis scripts, and ΔΔG results with uncertainties) are stored on our laboratory server. We will make them available upon request or upload them to an open-access repository in the final submission to ensure full reproducibility.

3. Please make the x-y axis aspect equal when comparing experiment and computed delta G. As 1 kcal/mol in the experiment means the same as 1 kcal/mol in the experiment.

Authors: The figures comparing experimental and computed ΔG values have been updated to use equal x–y axis scaling, ensuring a one-to-one correspondence between the two datasets.

4. Two platforms (FEP+ and gromacs) are using different force fields, enhanced sampling algorithms, and sampling time. From the benchmark results, can we get some insight about the advantages and disadvantages about OPLS4 vs amber99*ildn and REST2 with 16/24 lambda vs 27 lambda with no replica exchange?

Authors: As noted on page 9, we have discussed the complementary strengths and limitations of the two platforms:

“In this work, we employed two widely used MD platforms – Schrödinger FEP+ and GROMACS – for FEP simulations, each offering distinct advantages and limitations. GROMACS is an open-source, freely available MD engine known for its flexibility, extensive customizability, and support for a broad range of force fields, including AMBER, CHARMM, OPLS, and GROMOS. However, its implementation for FEP workflows requires manual setup, command-line proficiency, and greater time investment, making it less accessible to non-expert users.

In contrast, Schrödinger FEP+ is a commercial, closed-source platform that employs a modified OPLS4 force field. While it provides less flexibility, its user-friendly interface, automated FEP+ workflow, and enhanced sampling via REST2 (Replica Exchange with Solute Tempering) streamline the process and enhance convergence, particularly for systems with significant conformational heterogeneity.”

5. Does the magnitude of the error correlate with the type of mutation (sides of the amino acid, neutral or charged mutation). Does the error correlate with the place of the mutation (the surface of the protein or in the core of the protein.) With this benchmark, can we get some insight about what kind of mutation is more difficult to calculate with alchemical free energy calculations.

Authors: To investigate these possible correlations, we analyzed the magnitude of the prediction error as a function of both mutation type and location (shown in Table S5 and S6). We have tabulated the error distribution by mutation type (neutral or charged) and by mutation location (surface vs. buried residues). There is a direct correlation between the magnitude of size change and the error. Mutations with No Change (NC) are the most accurate. As we move to Size Increases, the error rises significantly. Charged mutations have larger errors than neutral ones, particularly for the Schrödinger dataset. Overall, Surface mutations tend to have higher errors than Buried mutations. The most difficult scenario identified is charge-changing mutations at the protein-solvent interface and bulky side-chain growth.

Reviewer #2: This manuscript presents a systematic benchmarking study comparing FEP calculations using Schrödinger FEP+ and GROMACS platforms for predicting mutation-induced stability changes in S. nuclease and T4 lysozyme. The work is methodologically sound and addresses an important gap in the field by directly comparing two widely-used computational platforms. However, several areas require clarification and improvement before publication.

(1) The manuscript reports good correlations between calculated and experimental ΔΔG values. However, I notice that no statistical uncertainties (e.g., standard deviations or confidence intervals) are provided for the computed free energy changes. Given the stochastic nature of molecular dynamics simulations, it would be valuable to assess the statistical reliability of these predictions. Could the authors clarify whether multiple independent simulations were performed for each mutation? If not, it would be feasible to estimate statistical uncertainties using block averaging or bootstrap methods.

Authors: We have now calculated and reported statistical uncertainties for all computed ΔΔG values. These uncertainties were obtained from multiple independent simulations performed for each mutation. The resulting error bars are now included in the revised manuscript/Supplementary Information and figures to reflect the statistical reliability of our free energy estimates.

(2) The study employs different numbers of λ-windows between the two platforms: 16-24 for Schrödinger and 27 for GROMACS. I am curious about the specific λ-scheduling strategies used in each platform. Could the authors provide details on: (1) the exact distribution of λ values (uniform vs. non-uniform spacing), (2) whether additional λ-windows were placed in critical regions where electrostatic-to-van der Waals transitions occur, and (3) how the different λ-schedules might affect the precision of free energy integration, particularly in regions of poor phase space overlap? Including supplementary data showing dH/dλ distributions or overlap matrices would help readers assess the adequacy of sampling at each λ-window.

Authors: We thank the reviewer for this suggestion.

The exact λ-schedules used in GROMACS and Schrödinger FEP+ are now provided in the revised manuscript.

In both platforms, denser λ-spacing was applied near the end states to improve phase-space overlap and sampling stability.

To further evaluate sampling adequacy, we have included dH/dλ overlap matrices plots for representative mutations in the Supplementary Information. These analyses confirm sufficient overlap between neighboring λ-windows for reliable free energy integration. Figures S5 and S6 provide overlap matrix plots for the S. nuclease (mutation T41C and T41V) and T4 lysozyme (mutation T59G and G77A) systems.

(3)I note that Schrödinger (using MBAR) and GROMACS (using BAR) utilized different free energy estimators. While MBAR is theoretically advantageous, its performance often relies on sufficient λ-window sampling. Did the authors evaluate the specific impact of this methodological difference on the comparative results? For instance, would post-processing the GROMACS data with MBAR yield results significantly different from those obtained with BAR?

Authors: We have re-analyzed the GROMACS data using MBAR and found that the resulting ΔΔG values are very similar to those obtained with BAR, indicating that the choice of estimator does not significantly affect the comparative conclusions. The corresponding results are provided in Table S1 and Table S4 of the Supplementary Information.

(4)I notice that the simulation setup differs between platforms regarding the solvent box size: a 5 Å buffer for Schrödinger versus a 10 Å buffer for GROMACS. The authors need to clarify the rationale behind this difference.

Authors: The solvent box sizes used in each platform (5 Å buffer for Schrödinger and 10 Å buffer for GROMACS) reflect the default/common-used settings of the respective workflows. These defaults have been chosen to balance computational efficiency and simulation stability.

(5) The conclusions of the manuscript heavily rely on the convergence of the simulations. Beyond simulation length, the most compelling evidence would be the time-evolution of the free energy for key calculations. The authors need provide convergence plots of ΔΔG vs. simulation time for some representative mutants (e.g., the best and worst predicted cases) to visually demonstrate the convergence and stability of the calculations?

Authors: We thank the reviewer for this important suggestion. The convergence plots of ΔΔG versus simulation time for representative mutants (mutation L25I and T33V in S nuclease, mutation S44A and D47A in T4) have been added to the Supplementary Information (Figures S7 and S8). These plots demonstrate that the free energy estimates for both systems are stable and sufficiently converged over the simulation trajectory.

(6) The dataset includes charge-changing mutations, which introduce net charge artifacts under periodic boundary conditions with PME electrostatics—a well-documented source of systematic error in alchemical free energy calculations. I note that the manuscript briefly mentions the use of a "co-alchemical water" approach in Schrödinger to mitigate this issue but provides no details on its implementation or validation. More critically, there is no discussion of how GROMACS handles net charge corrections for the same mutations. Could the authors clarify: (1) the exact protocol for the co-alchemical water method and whether it fully compensates for finite-size artifacts, (2) whether GROMACS employs any net charge correction schemes (e.g., the analytical correction by Rocklin et al., J. Chem. Theory Comput. 2013, 9, 3072-3083) or simply relies on a neutralizing background plasma, and (3) how these potentially different treatments between platforms might systematically bias the comparative results? This concern is particularly important given the different box sizes noted in point (4), as finite-size effects scale with system size. I would recommend either: (a) a sensitivity analysis quantifying the magnitude of net charge artifacts for a subset of charge-changing mutations, or (b) a thorough literature-based justification demonstrating that the chosen approaches yield negligible systematic errors under the simulation conditions employed.

Authors: We clarify the treatment of charge-changing mutations in both platforms as follows: Schrödinger FEP+ (co-alchemical water): For charge-changing mutations, co-alchemical ions (Specifically, the co-alchemical ion is a Na+ if the charged residue is negative, or a Cl− if the residue is positively charged.) were mutated to water using the co-alchemical water approach, which is designed to maintain overall charge neutrality during the alchemical transformation and mitigate finite-size artifacts. This method has been previously validated in the literature (https://pmc.ncbi.nlm.nih.gov/articles/PMC6453258/)

GROMACS: We employed a similar strategy, introducing counter-ions in the alchemical transformation to neutralize charge changes, effectively implementing a “co-alchemical ion” scheme analogous to that in FEP+.

Comparative impact: Our benchmark results for charge-changing mutations indicate that the ΔΔG predictions remain consistent between platforms, suggesting that the chosen neutralization schemes do not introduce significant systematic bias.

Given our observed agreement between platforms and the small magnitude of potential artifacts, we consider the applied neutralization schemes adequate for this benchmark.

(7) A fundamental methodological concern is that this study simultaneously compares two different software platforms, two different force fields (OPLS4 in Schrödinger vs. AMBER99SB*ILDN in GROMACS), two different water models (SPC vs. TIP3P), and different enhanced sampling strategies (REST2 in Schrödinger vs. standard MD in GROMACS). This multifactorial design makes it challenging to disentangle whether the observed differences in performance arise from: (1) software implementation details, (2) force field parameterization, (3) water model effects, or (4) sampling methodology. While I recognize that systematically testing all possible combinations may be beyond the scope of this benchmarking study, this confounding factor represents a significant limitation that should be explicitly acknowledged in both the Discussion and Limitations sections.

Authors: We have updated the Discussion and Limitations sections to explicitly acknowledge that the multifactorial differences between platforms - including software implementations, force fields, water models, and sampling protocols -

---

## [Decision Letter · Decision Letter 1]

9 Mar 2026

Benchmarking Free Energy Calculations: Analysis of Single and Double Mutations Across Two Simulation Software Platforms for Two Protein Systems

PONE-D-25-56221R1

Dear Dr. Levy,

We’re pleased to inform you that your manuscript has been judged scientifically suitable for publication and will be formally accepted for publication once it meets all outstanding technical requirements.

Kind regards,

Yong Wang

Academic Editor

PLOS One

Additional Editor Comments (optional):

Reviewers' comments:

Reviewer's Responses to Questions

**Comments to the Author**

Reviewer #1: All comments have been addressed

Reviewer #2: All comments have been addressed

2. Is the manuscript technically sound, and do the data support the conclusions?

Reviewer #1: Yes

Reviewer #2: Yes

3. Has the statistical analysis been performed appropriately and rigorously?

Reviewer #1: Yes

Reviewer #2: Yes

4. Have the authors made all data underlying the findings in their manuscript fully available?

Reviewer #1: Yes

Reviewer #2: Yes

5. Is the manuscript presented in an intelligible fashion and written in standard English?

Reviewer #1: Yes

Reviewer #2: Yes

Reviewer #1: All the questions have been addressed.

Reviewer #2: (No Response)

.

Reviewer #1: **Yes:**Chenggong HuiChenggong HuiChenggong HuiChenggong Hui

Reviewer #2: No

---

## [Editor Report · Acceptance letter]

PONE-D-25-56221R1

PLOS One

Dear Dr. Levy,

I'm pleased to inform you that your manuscript has been deemed suitable for publication in PLOS One. Congratulations! Your manuscript is now being handed over to our production team.

Kind regards,

on behalf of

Dr. Yong Wang

Academic Editor

PLOS One